# Modeling treatment of osteoarthritis with standard therapy and senolytic drugs

**Nourridine Siewe**[1]*, **Avner Friedman**[2]

**1** School of Mathematics and Statistics, Rochester Institute of Technology, Rochester, New York, United States of America, **2** Department of Mathematics, The Ohio State University, Columbus, Ohio, United States of America

☙ These authors contributed equally to this work.

* nxssma@rit.edu

**Citation:** Siewe N, Friedman A 2025 Modeling treatment of osteoarthritis with standard therapy and senolytic drugs. PLoS One 20(9): e0332763. https://doi.org/10.1371/journal.pone.0332763

**Data availability statement:** All relevant data are within the manuscript.

## Abstract

Osteoarthritis (OA), the most common form of joint disease, involves the progressive degradation of articular cartilage and is a major cause of chronic disability in aging populations. Since OA is associated with severe deficiency of collagen type II, clinical trials considered treatment of OA by injection with undenatured collagen type II (UC-II). Recent studies consider also injection of senolytic drugs, like fisetin, that eliminates senescent chondrocytes in aging patients, to reduce the negative effect of these senescent cells on cartilage structure. In this paper we develop a mathematical model of OA for men and, separately, for women, and use the model to assess the efficacy of treatment by UC-II and by fisetin, alone or in combination. Our computations show the benefits of starting treatment early. They also show that although the effect of treatment by fisetin on slowing the progression of OA is much smaller compared to UC-II treatment, its effect in combination with UC-II is significantly increased.

## Introduction

Arthritis is a term commonly used to mean any disorder that affects the joint; symptoms include join pain and stiffness. Osteoarthritis (OA) is the most common joint disease affecting 500 million people worldwide [1]. The disease wears down the protective cartilage that cushions the end of bone of a joint, and the damage cannot be reversed [2]. Cartilage is a spongy tissue containing up to 80% water, chondrocyte cells, and collagen produced by the chondrocytes; 90–95% of the collagen are collagen type II [3]. Chondrocytes receive nutrients and oxygen by diffusion from the synovial fluid and from the adjacent subchondral bone.

Articular cartilages are a class of cartilages with specific arrangement of chondrocytes and collagen fibers; they is found in the knee and hip joints, and in finger joints. Meniscus is the articular cartilage in the knee joint, below the synovial fluid, that functions as shock absorber.

Cellular senescence is a permanent arrest of normal cell cycle, while maintaining cell viability. Senescence cells secrete senescence-associated secretory phenotype (SASP) that include inflammatory proteins and MMP1, MMP13 [4]; the level of MMP13 is highly expressed in OA cartilage [5].

**Funding:** Research reported in this publication was supported by the National Institute Of General Medical Sciences of the National Institutes of Health under Award Number R16GM154782. The content is solely the responsibility of the authors and does not necessarily represent the official views of the National Institutes of Health. The funders had no role in study design, data collection and analysis, decision to publish, or preparation of the manuscript.

**Competing interests:** The authors have declared that no competing interests exist.

Cellular senescence is the primary hallmark of aging. OA in age-related cartilage degenerative disease is most commonly caused by chondrocyte senescence [6]. OA of knee joints affect most adults 65 and older, 33% of men and 42% of women in the United States; and knee arthritis is the most common cause of disability affecting 1 in 5 adults in the United States [7].

Arthritis can be inflammatory or non-inflammatory. In most of the population that develop OA after age 50, the disease is associated with age and it is non-inflammatory [8]. In inflammatory OA, the body immune system is causing the inflammation, and there remain considerable gaps in understanding its role in disease progression and how best to target inflammatory response for therapeutic intervention [9].

In this paper we developed a model of non-inflammatory arthritis for population over 50.

Chondrocyte senescence results in decline of chondrocyte cells; for example, there is a 30% fall of chondrocytes in articular cartilage of the hip joint between ages 30 to 70 [10]. Recent studies reveal the presence of mesenchymal stem cells (MSC) in the synovial fluid, synovial membrane, and articular cartilage [11]. Although MSC may sustain the source of chondrocytes or contribute to senescence in OA, because of lack of data we do not include MSC in our model, and treat the source of chondrocytes as a fixed parameter.

Chondrocyte hypertrophy is a process by which chondrocyte cells undergo 10 to 20-fold enlargement [12]. Hypertrophic chondrocytes are master regulators of endochondral ossification, i.e., chondrocyte-to-osteoblast transdifferentiation [12]. Hyertrophic chondrocytes lie near the boundary between cartilage and bone, and secrete factors that promote cartilage to bone transition [4]; they play an essential role in bone formation in development [4,5]. Chondrocytes transition to hypertrophic state in fracture healing [4,5].

Calcified cartilage is a layer of hard connective tissue between articular cartilage and sub-chondral bone; the calcification results from deposition of insoluble calcium salt in the bone matrix [13]. Calcium crystals in calcified cartilage play a role in pathogenesis of OA [14], Hypertrophic chondrocytes serve a critical intermediate in cartilage calcification and in OA lesions [15,16]; they mediate calcified cartilage through membrane bound enzymes and secreted matrix vesicles [17]. Pathological calcification is the hallmark of OA; calcification can be observed both at cartilage surface and in deeper levels [18]. Meniscal calcification is a predisposing factor for cartilage lesions, and is a target for disease modifying drugs of OA [19].

Collagen type II suppresses chondrocyte hypertrophy and OA progression [20], and degradation of collagen type II is associated with articular cartilage lesions [21]. In OA, the highly expressed MMP13 degrades the ECM structure, which results in decreased of collagen type II [22], hence it is an attractive target for treatment of OA [23].

Histological assessment of OA in mouse is given in [24]; in particular, grade 4 of OA is when lesion reaches the calcified cartilage for 25–50% of the quadrant width, grade 5 for 50–75%, and grade 6 when the width is above 75%.

Several mathematical and theoretical models have been proposed over the years to better understand the biomechanics and pathology of articular cartilage. A foundational study by Hayes et al. (1972) developed a mathematical model for indentation tests of articular cartilage, providing early insight into its mechanical behavior [25]. Later, Parra et al. (2011) introduced a model that quantifies the risk of cartilage failure by treating the tissue as an elastic material subjected to cyclic loading, highlighting the role of mechanical fatigue [26]. Kapitanov et al. (2016) advanced the field by proposing a PDE-based model that explicitly couples mechanical stress with biological processes in cartilage lesion formation [27]. In 2019, Campbell et al. focused on the regenerative aspects, modeling the effects of growth factors on cartilage repair following cell implantation [28], while Cope et al. critically reviewed existing osteoarthritis (OA) models and emphasized the need for a unified, physiologically accurate model [29].

Eschweiler et al. (2021) provided a comprehensive overview of cartilage biomechanics, summarizing key concepts relevant to both modeling and clinical application [30]. Most recently, Owida et al. (2022) reviewed the advancements in biomimetic strategies for cartilage tissue engineering, discussing their implications for future therapeutic approaches [31].

In this paper we develop a mathematical model of OA that includes chondrocytes ($C$), senescent chondrocytes ($C_s$), hypertrophic chondrocytes ($C_h$), calcified cartilage ($C_c$), MMP13 ($M_P$), collagen II ($L_2$) and undenatured collagen type II ($U_2$). (Undenatured collagen type II is a specific form of collagen II derived from chicken stertum cartilage. This form of collagen reduces joint pains associated with immune response, when injected to OA patients.) The model includes treatment of OA by $U_2$ injection and, separately, by senolytic drug fisetin. We demonstrate how the model can qualitatively replicate reported clinical studies with $U_2$ injection. We then conduct clinical studies *in-silico* with these drugs, separately and in combination, and demonstrate the synergy of the combination.

## 1 Mathematical model

Table 1 lists the model variables in units of g/cm$^3$. Fig 1 shows the interactions among the model variables. Chondrocytes ($C$) can become senescent ($C_s$) and senescent cells secrete MMP13 ($M_P$). Chondrocytes can become hyperthophic ($C_h$), a process blocked by collagen type II ($L_2$). $M_P$ blocks $L_2$, which results in abnormal growth in $C_h$ and formation of calcified cartilage ($C_c$).

Fig 2 shows normal knee joint anatomy, and a simplified geometry used in the model.

The mathematical model is based on the network in Fig 1, and it takes place in the strip $\Omega = \{0 \leq x \leq h\}$ of the simplified geometry shown in Fig 2. We consider aging population between 50 and 90 years.

**Table 1. Variables of the model.** Densities and concentrations are in units of g/cm$^3$.

| Descriptions | Variables | Descriptions | Variables |
|---|---|---|---|
| chondrocyte cells | $C$ | senescence chondrocytes | $C_s$ |
| hypertrophic chondrocytes | $C_h$ | calcified cartilage | $C_c$ |
| MMP13 | $M_P$ | collagen type II | $L_2$ |
| senolytic drug fisetin | $D$ | undenatured collagen type II | $U_2$ |

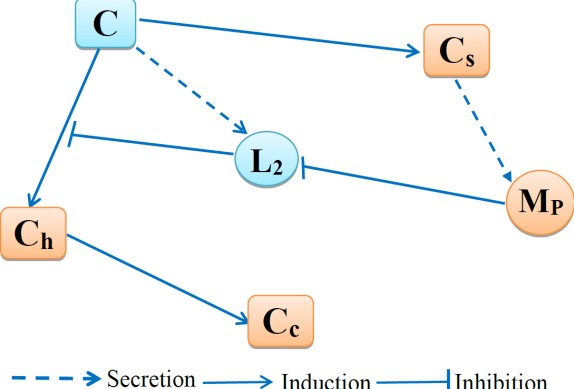

**Fig 1. Interactions among cells and proteins.** $C$ = chondrocytes, $C_s$ = senescent chondrocytes, $C_h$ = hypertrophic chondrocytes, $C_c$ = calcified cartilage, $M_P$ = MMP13, $L_2$ = collagen type II.

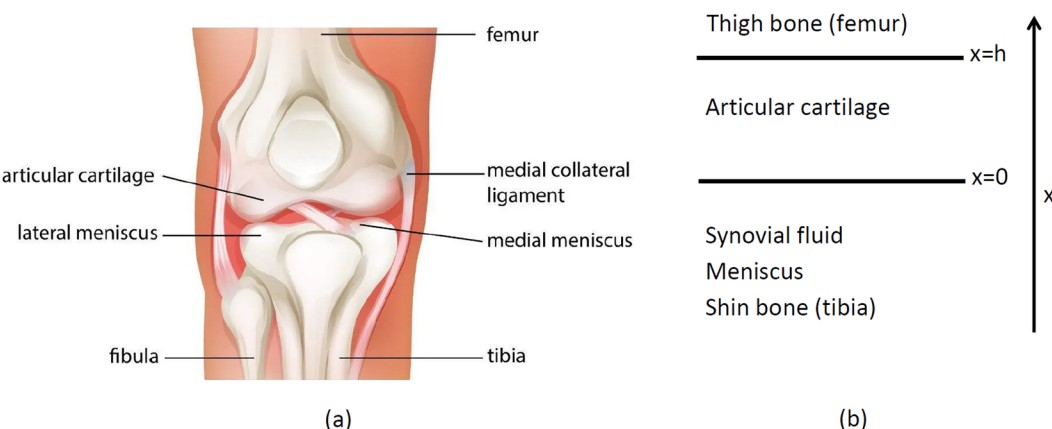

**Fig 2. (a) Knee joint anatomy (adapted from [32]).** (b) simplified geometry of articular cartilage.

## Model equations in males and females

In [33], a meta-analysis of animal models is conducted, highlighting that during the post-menopausal period, as estrogen levels decrease, there is a corresponding reduction in collagen type II in cartilage. The study in [34] further demonstrates that MMP-13 levels also increase in cartilage in murine subjects following menopause. The impact of estrogen depletion during postmenopause may help explain why the global prevalence of OA in women over the age of 60 is nearly twice as high as in men [34].

Based on Fig 1 and [33,34], we write the model equations as follows.

**Equations for cells ($C_s$, $C$ and $C_h$) and calcified cartilage ($C_c$).** We denote the source of chondrocytes in health by $A$, and write the following equations:

$$\frac{\partial C}{\partial t} - \delta_C \frac{\partial^2 C}{\partial x^2} = A - \lambda_{CC_s} C - \lambda_{CC_h} C \frac{1}{1 + L_2/K_{L_2}} - \mu_C C \tag{1}$$

$$\frac{\partial C_s}{\partial t} - \delta_C \frac{\partial^2 C_s}{\partial x^2} = \lambda_{CC_s} C - \mu_{C_s} C_s \tag{2}$$

$$\frac{\partial C_h}{\partial t} - \delta_C \frac{\partial^2 C_h}{\partial x^2} = \lambda_{CC_h} C \frac{1}{1 + L_2/K_{L_2}} - \lambda_{C_h C_c} \frac{\partial C_h}{\partial t} - \mu_{C_h} C_h \tag{3}$$

$$\frac{\partial C_c}{\partial t} - \delta_{C_c} \frac{\partial^2 C_c}{\partial x^2} = \lambda_{C_h C_c} \frac{\partial C_h}{\partial t}. \tag{4}$$

Here, $\lambda_{CC_s}$ is the rate at which chondrocytes become senescent, and $\lambda_{CC_h}/(1 + L_2/K_{L_2})$ is the rate at which chondrocytes become hypertrophic under blockage by collagen type II; $K_{L_2}$ is the half-saturation of $L_2$. The term $\lambda_{C_h C_c} \partial C_h/\partial t$ models calcified cartilage formation induced by chondrocyte hypertrophy dynamics, where $C_c$ arises not from a direct $C_h \rightarrow C_c$ transition but as a consequence of $C_h$ activity [16]. The parameters $\mu_C$, $\mu_{C_s}$ and $\mu_{C_h}$ are death rates, and $\lambda_{C_h C_c}$ is the rate of increase of calcified cartilage. The immune cells clear senescent cells [35], and since the strength of the immune system declines with age [36], $\mu_{C_s} = \mu_{C_s}(t)$ is an increasing function of $t$; we take $\mu_{C_s}(t)$ a putative increasing function

$$\mu_{C_s}(t) = \frac{\mu_{C_s}(0)}{1 + t/(90 \text{ years})} \text{ d}^{-1}. \tag{5}$$

**Equations for proteins.**

$$\frac{\partial M_P}{\partial t} - \delta_{M_P}\frac{\partial^2 M_P}{\partial x^2} = \begin{cases} \lambda_{C_s M_P}C_s - \mu_{L_2 M_P}M_P L_2 - \mu_{M_P}M_P, & \text{for males,} \\ \lambda_{C_s M_P}C_s - \mu_{L_2 M_P}M_P L_2 - \mu_{M_P}M_P + d_{pM_P}(t)M_P, & \text{for females,} \end{cases} \tag{6}$$

$$\frac{\partial L_2}{\partial t} - \delta_{L_2}\frac{\partial^2 L_2}{\partial x^2} = \begin{cases} \lambda_{CL_2}C - \mu_{M_P L_2}L_2 M_P - \mu_{L_2}L_2, & \text{for males,} \\ \lambda_{CL_2}C - \mu_{M_P L_2}L_2 M_P - \mu_{L_2}L_2 - d_{pL_2}(t)L_2, & \text{for females.} \end{cases} \tag{7}$$

The parameter $\lambda_{C_s M_P}$ denotes the production rate of MMP13 by senescent chondrocytes, and $\lambda_{CL_2}$ represents the secretion rate of collagen type II by chondrocytes. The parameters $\mu_{L_2 M_P}$ and $\mu_{M_P L_2}$ describe the mutual elimination rates between collagen type II and MMP13. Degradation rates are $\mu_{M_P}$ for MMP13 and $\mu_{L_2}$ for collagen type II, while $d_{pM_P}(t)$ and $d_{pL_2}(t)$ are postmenopausal monotone decreasing functions, with $d_{pM_P}(0) > 0$, $d_{pL_2}(0) > 0$ (age 50) and $d_{pM_P}(t) \sim 0$, $d_{pL_2}(t) \sim 0$ if $t \geq 10$ years (age $\geq 60$). We take

$$d_{pX}(t) = \frac{\alpha_{wX}}{1 + t^2/T_{wX}^2}, \quad X = M_P, L_2, \tag{8}$$

where $\alpha_{wX}$ and $T_{wX}$ are postmenopausal effect and time-scale postmenopausal effect of species $X$.

**Equations with drugs.**

$$\frac{\partial C_s}{\partial t} - \delta_C\frac{\partial^2 C_s}{\partial x^2} = \lambda_{CC_s}C - \mu_{C_s}C_s - \mu_{C_s D}C_s D \tag{9}$$

$$\frac{\partial M_P}{\partial t} - \delta_{M_P}\frac{\partial^2 M_P}{\partial x^2} = \begin{cases} \lambda_{C_s M_P}C_s\dfrac{1}{1 + \alpha_D D(t)} - \mu_{L_2 M_P}M_P L_2 - \mu_{M_P}M_P, & \text{for males,} \\ \lambda_{C_s M_P}C_s\dfrac{1}{1 + \alpha_D D(t)} - \mu_{L_2 M_P}M_P L_2 - \mu_{M_P}M_P + d_{pM_P}(t)M_P, & \text{for females,} \end{cases} \tag{10}$$

$$\frac{\partial L_2}{\partial t} - \delta_{L_2}\frac{\partial^2 L_2}{\partial x^2} = \begin{cases} \alpha_{U_2}U_2(t) + \lambda_{CL_2}C - \mu_{M_P L_2}L_2 M_P - \mu_{L_2}L_2, & \text{for males,} \\ \alpha_{U_2}U_2(t) + \lambda_{CL_2}C - \mu_{M_P L_2}L_2 M_P - \mu_{L_2}L_2 - d_{pL_2}(t)L_2, & \text{for females.} \end{cases} \tag{11}$$

The parameter $\mu_{C_s D}$ denotes the rate at which senescent chondrocytes ($C_s$) are eliminated due to $D$. The constants $\alpha_D$ and $\alpha_{U_2}$ scale the effects of $D$ and $U_2$, respectively.

The equations for the senolytic drug $D$, and UC-II ($U_2$) ($L_2$ injection) take the form

$$\frac{\partial D}{\partial t} - \delta_D\frac{\partial^2 D}{\partial x^2} = \gamma_D h_D(t) - \mu_{DC_s}DC_s - \omega_D D \tag{12}$$

$$U_2(t) = \gamma_{U_2}h_{U_2}(t), \tag{13}$$

where $\mu_{DC_s}$ is the rate of lost of $D$ while eliminating $C_s$, $\omega_D$ is the washout rate of $D$, and

$$h_{U_2}(t) = \begin{cases} 1, & \text{if the drug } U_2 \text{ is given (daily) at time } t \\ 0, & \text{otherwise;} \end{cases}$$

if $D$ is administered only at days $t_0$ and $t_1$, then during the period $t_0 \leq t \leq t_2$ (where $t_2 > t_1$),

$$h_D(t) = \begin{cases} e^{-\mu_D t}, & \text{for } t_0 \leq t \leq t_1 \\ e^{-\mu_D t} + e^{-\mu_D(t-t_1)}, & \text{for } t_1 < t < t_2, \end{cases}$$

where $\mu_D$ is the degradation rate of the drug $D$.

The parameters $\alpha_D$ and $\alpha_{U_2}$ are constants representing the effects of the respective drugs, $\omega_D$ is the washout rate of $D$; $t_0$, $t_1$ and $t_2$ are time points for delivering of $D$, and the parameters $\gamma_D$ and $\gamma_{U_2}$ are the doses of the drugs.

**Boundary conditions.** We assume no-flux boundary conditions for all the viable cells:

$$\frac{\partial X}{\partial x} = 0 \text{ on } x = 0 \text{ and } x = h \text{ for cells } X = C, C_s, C_h, \tag{14}$$

and

$$\frac{\partial X}{\partial n} + \alpha X = 0 \text{ on } x = 0 \text{ and } x = h \text{ for proteins } X = L_2, M_P \text{ and drug } D, \tag{15}$$

for some $\alpha > 0$, where $\partial/\partial n$ is the outward normal derivative.

Eq (14) means that cells $X$ do not flow in or out of the cartilage, while Eq (15) means that proteins $X$ leak out of the cartilage at rate $\alpha$.

We take initial conditions in units of g/cm$^3$ as follows:

$$C(0) = 3 \times 10^{-2}, \ C_s(0) = 4.5 \times 10^{-3}, \ C_h(0) = 2.7 \times 10^{-4}, \ C_c(0) = 0,$$
$$M_P(0) = 2.6 \times 10^{-7}, \ L_2(0) = 1.7 \times 10^{-4}. \tag{16}$$

Decrease in the growth of $C_c$ means slower growth in calcified cartilage, which results in slower growth in the progression of OA [24]. We accordingly define the efficacy of the drug $X$ for a treatment beginning at $t_0$ and ending at $t$ by the relative reduction in calcified cartilage:

$$\text{Efficacy} = \frac{C_c(t; t_0, \text{no drug}) - C_c(t; t_0, X)}{C_c(t; t_0, \text{no drug})} (\times 100)\%. \tag{17}$$

This definition is the same for men and women. But in considering clinical studies with mixed population of men and women, we define E-efficacy as follows:

$$\text{E-efficacy} = \frac{\text{\# of males}}{\text{\# of males and females}} \times (\text{Efficacy for males}) \tag{18}$$
$$+ \frac{\text{\# of females}}{\text{\# of males and females}} \times (\text{Efficacy for females})$$

In Fig 3, we display the profiles of all the model variables for both males and females. From the non-linear system (1)–(7) it is difficult to deduce the relative differences between men and women. But, since $L_2$ is decreasing faster in women, both $C$ and $C_s$ profiles are larger for men, as they should be by Eqs (1)–(2). However, since $C$ and $L_2$ are both larger for men, it is not clear if $C/(1 + L_2/K_{L_2})$ is generally larger or smaller for men than for women, which would affect the profiles of $C_h$ and $C_c$ of men versus women. Fig 3 shows, that the profiles of $C_h$ and $C_c$ are larger for women, indicating that women are more susceptible to OA. The profile of $M_P$ reflects a sharp postmenopausal effect, where $M_P$ first sharply increases and decreases, and then, after age 70, slowly increases.

## 2 Drugs

### Undenatured type II collagen (UC-II)

**Treatment 1 (T1), from [37]**

30 males and 22 females, aged between 40 and 75 years, received either UC-II (40 mg/day) or placebo for 90 days. At day 90, the WOMAC score of pain level for the group receiving

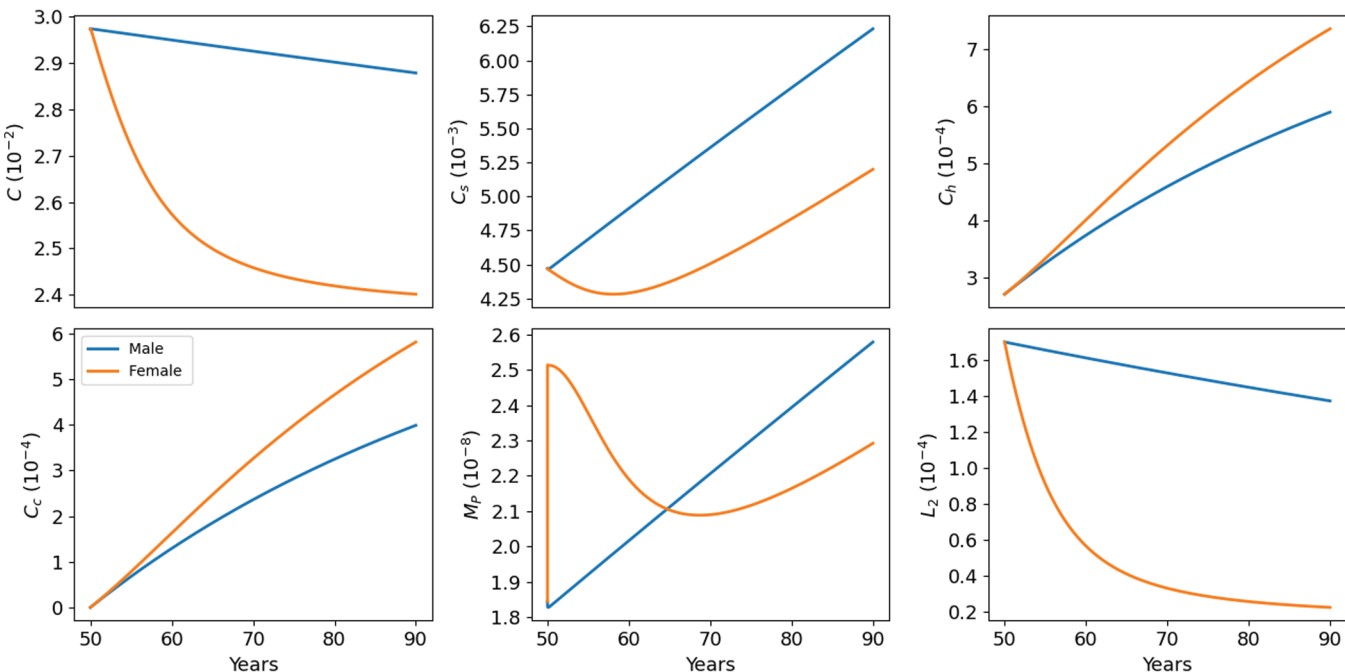

**Fig 3. Profiles of the average density over the strip $\Omega = \{0 \leq x \leq h\}$ of the model variables for males and females in the control case (no drugs).**

UC-II was 33% reduced from the group that received placebo. In our simulations, we take the average age of all participants at the clinical study to be 63 years.

**Treatment 2 (T2), from [38]**

89 males and 97 females, aged between 40 and 75 years, received either UC-II (40 mg/day) or placebo for 90 days. At day 90, the WOMAC score of pain level for the group receiving UC-II was 32% reduced from the group that received placebo. In our simulations, we take the average age of all participants at the clinical study to be 63 years.

**Treatment 3 (T3), from [39]**

36 males and 69 females, aged between 60 and 80 years, received either UC-II (40 mg/day) or placebo for 90 days. At day 90, the WOMAC score of pain level for the group receiving UC-II was 22.16% reduced from the group that received placebo. In our simulations, we take the average age of all participants at the clinical study to be 70 years.

**Treatment 4 (T4), from [40]**

39 participants, all females, aged between 40 and 80 years, received either UC-II (40 mg/day) or placebo for 90 days. At day 90, the WOMAC score of pain level for the group receiving UC-II was 25.5% reduced from the group that received placebo. In our simulations, we take the average age of all participants at the clinical study to be 65 years.

## Model's simulations of Treatments T1–T4

In Figs 4–7 we used the model equations to simulate the treatments T1–T4, and in the last panel of each figure we computed the E-efficacy of the treatment. Table 2 summarizes the data and experiment results of each treatment, and also includes (in the last column) the E-efficacy of the treatment computed by model's simulations. The protocol of treatment (amount and duration of the drug) was the same in all treatments. Treatment efficacy and treatment-associated pain are two different consequences of a treatment. Note that the E-efficacy (from

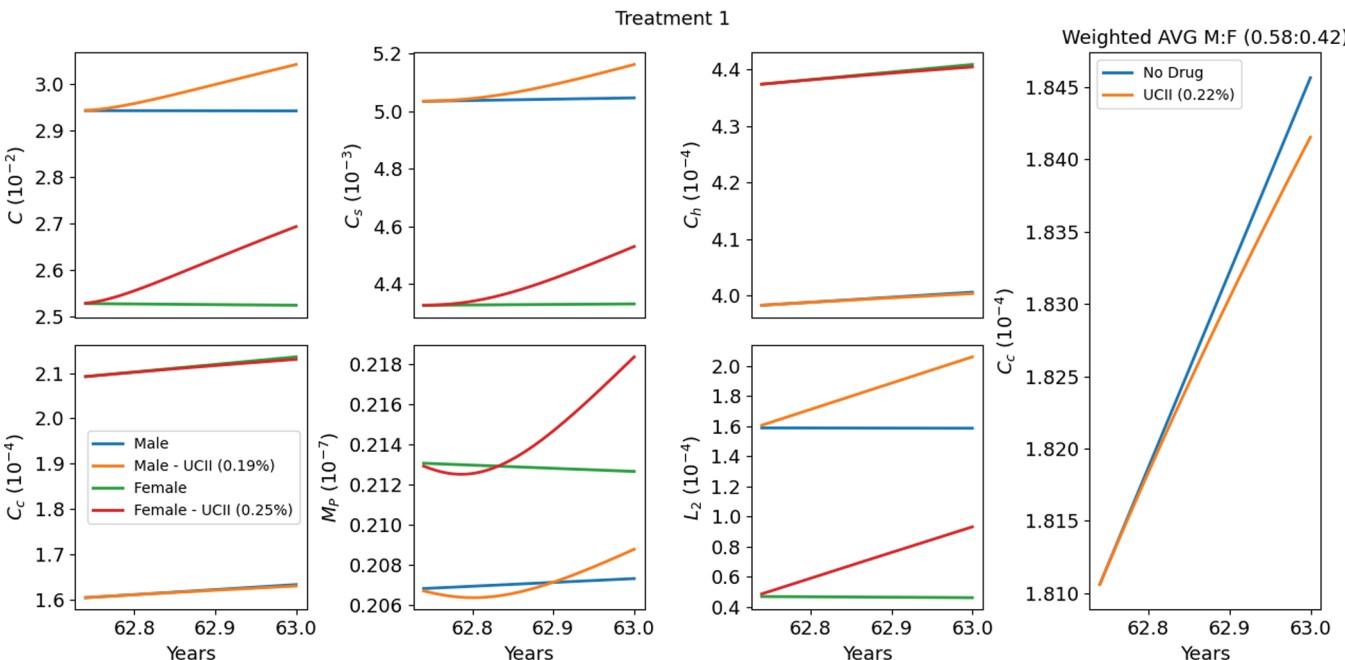

**Fig 4. Replicating Treatment 1 [37].** Profiles of the variables with and without the drugs. The extreme right panel provides a zoomed-in view of the profile of the weighted average of $C_c$ around the treatment period (90 days of treatment between years 62.75–63) relative to the male-to-female ratio. The E-efficacy of the treatment is 0.22%.

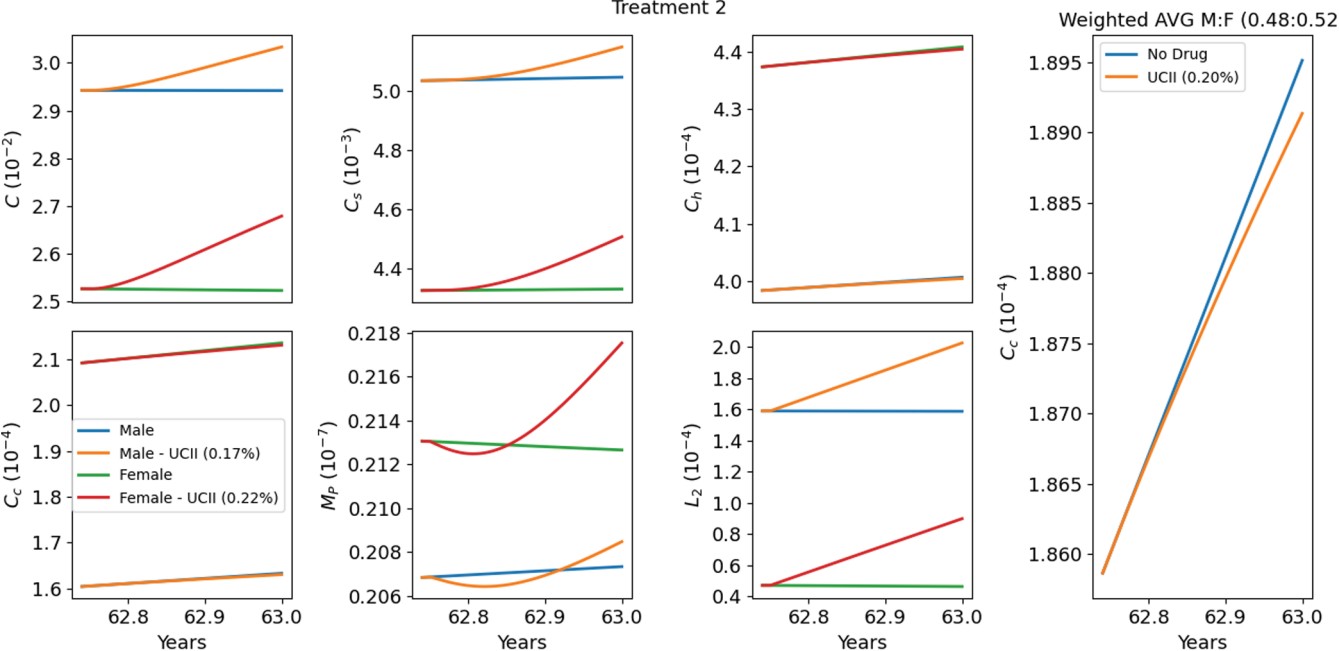

**Fig 5. Replicating Treatment 2 [38].** Profiles of the variables with and without the drugs. The extreme right panel provides a zoomed-in view of the profile of the weighted average of $C_c$ around the treatment period (90 days of treatment between years 62.75–63) relative to the male-to-female ratio. The E-efficacy of the treatment is 0.20%.

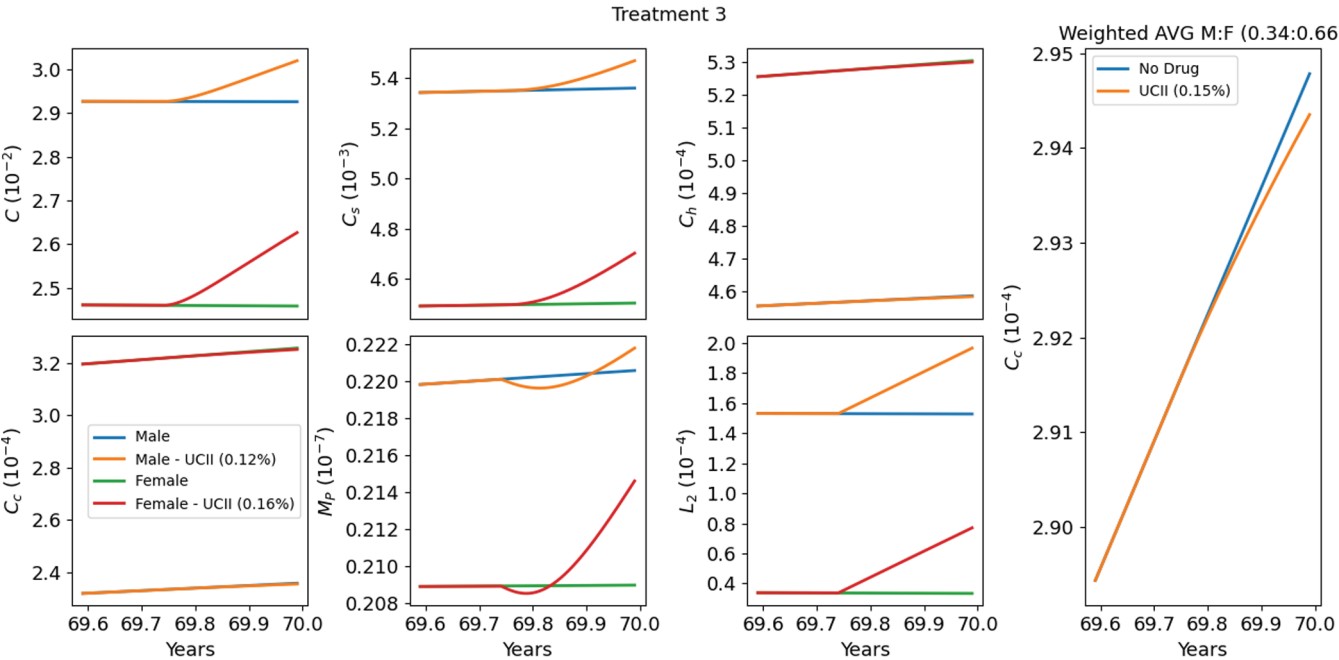

**Fig 6. Replicating Treatment 3 [39].** Profiles of the variables with and without the drugs. The extreme right panel provides a zoomed-in view of the profile of the weighted average of $C_c$ around the treatment period (90 days of treatment between years 69.75–70) relative to the male-to-female ratio. The E-efficacy of the treatment is 0.15%.

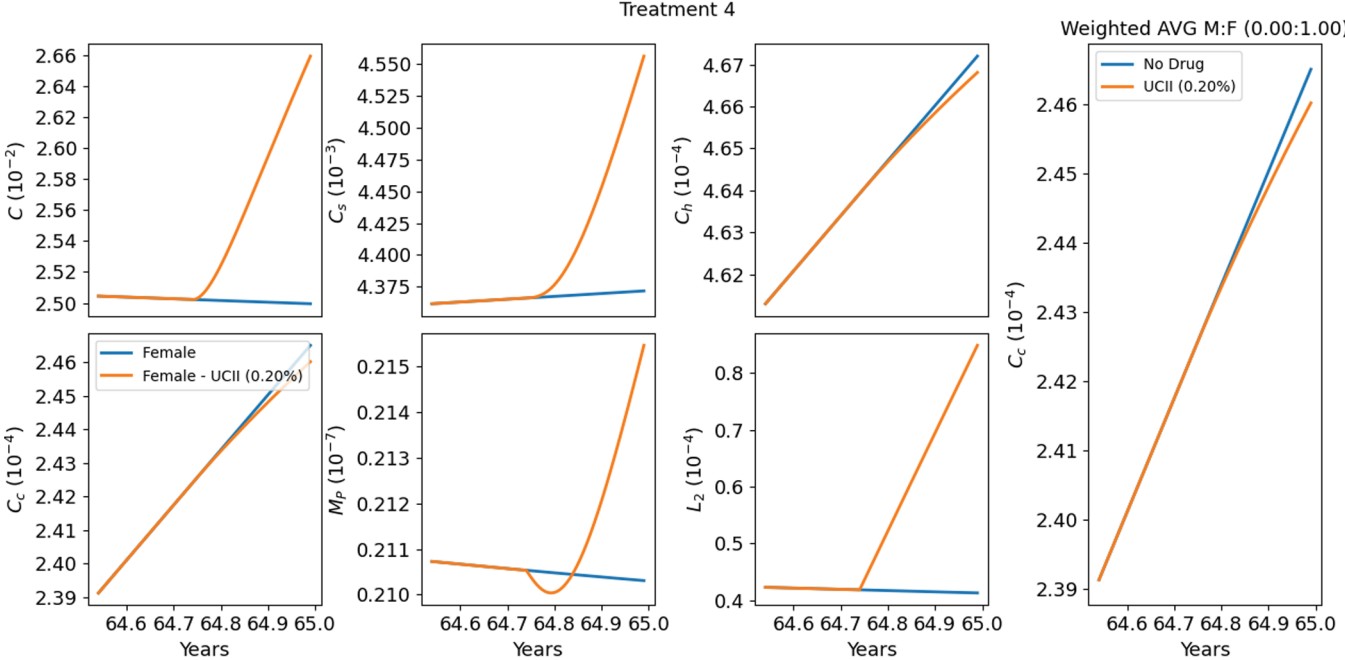

**Fig 7. Replicating Treatment 4 [40].** Profiles of the variables with and without the drugs. The extreme right panel provides a zoomed-in view of the profile of the weighted average of $C_c$ around the treatment period (90 days of treatment between years 64.75–65) relative to the male-to-female ratio. The E-efficacy of the treatment is 0.20%.

**Table 2. Summary of the comparison between model simulations and actual WOMAC score reports.**

|  | Source | M/F | AVG age | WOMAC reduction | E-efficacy (from model) | Figure |
|---|---|---|---|---|---|---|
| T3 | [39] | 36/69 | 70 | 22.16% | 0.15% | Fig 6 |
| T4 | [40] | 0/39 | 65 | 25.5% | 0.20% | Fig 7 |
| T2 | [38] | 89/97 | 63 | 32% | 0.20% | Fig 5 |
| T1 | [37] | 30/22 | 63 | 33% | 0.22% | Fig 4 |

the model) is very small (0.15–0.22%), because the treatment was taken over a relatively very short time, 90 days. On the other hand, the pain level was significantly reduced (22.16–33%). What is interesting to note is that the E-efficacy is positively correlated to WOMAC reduction: E-efficacy increases when WOMAC reduction increases. On the other hand WOMAC is positively correlated to the biological marker CTX-II, which has been shown to be associated with the severity of OA [41]. Hence the E-efficacy of treatment with UC-II, as derived by the model, is positively correlated to reduction in the severity of OA.

## Senolytics

**Clinical trials with UBX0101.** Clinical trials with senolytic molecule UBX0101 passed phase I, but failed in phase II [42].

**Clinical Trials with senolytic drug fisetin.** A total of 30 male and 44 female participants, aged between 40 and 80 years, were enrolled in the study [43]. Participants received either fisetin (100 mg/day) or a placebo for two consecutive days, followed by 28-day break, and then an additional two consecutive days of treatment. The primary outcome of the trial was assessment of treatment-emergent adverse events. Secondary outcome, assessed at multiple time points, WOMAC score, patient-reported outcomes (PRO) for knee pain, and levels of proinflammatory markers associated with cellular senescence. The results demonstrated that the study successfully met the primary objective of the clinical trial, with no serious side effects reported over a period of more than 12 months. According to the treatment protocol, the sparingly administered drug was not expected to have a significant impact on slowing the progression of OA. Nevertheless, notable improvements were observed after six months in four key physical function tests related to the study knee; see Table 3. This suggests that cartilage degeneration was slowed during approximately six months. Based on these findings, we conclude that fisetin shows some promise, and may merit advancement to the next phase of clinical trials.

**Table 3. Outcome measures from clinical trials with fisetin in [43].**

| Outcome Measures (since treatment initiation) | Placebo | Fisetin |
|---|---|---|
| Proinflammatory markers associated with senescent factors (0.5, 1.5, 6, 12 months) | 10.5, 6.4, 12.0, 8.8 | 14.0, 8.9, 32.5, 9.2 |
| Change in physical functions of the study knee (Time-up-and-go Test, 6, 12 months) | 6.09, 6.0 | 6.8, 5.93 |
| Change in physical functions of the study knee (Fast 4-meter Walk, 6, 12 months) | 21.7, 21.8 | 22.1, 21.7 |
| Change in physical functions of the study knee (LEK, 6, 12 months) | 0.182, 0.180 | 0.204, 0.204 |
| Change in physical functions of the study knee (Stair-Climbing Test, 6, 12 months) | 8.66, 8.78 | 8.88, 8.66 |
| Change in muscle strength (Isokinetic Dynamometry, 6, 12 months) | 83.1, 85.0 | 81.2, 81.2 |
| WOMAC (6, 12, 18 months) | 17.1, 15.4, 17.4 | 15.2, 19.6, 17.6 |
| Change in the quality of articular cartilage in the study knee with quantitative magnetic resonance imaging (MRI, 6, 12 months) | 150.3, 144.8 | 149.6, 156.1 |

## Clinical studies *in-silico*

Based on clinical studies with UC-II in T1–T4 and on clinical trial phase 1 with fisetin, we use our model to compute the E-efficacy of each of these treatments alone, and in combination.

In these studies we do not use the standard clinical trials schedule of 3 month on/off but 6 month for UC-II as in clinical studies T1–T4 [37–40] and schedules for fisetin based on clinical trials in [44].

In Fig 8, we present a simulation of treatment outcomes using various combinations of UC-II and fisetin, evaluating E-efficacy at age 63 for patients who began treatment at age 62. UC-II is administered daily in cycles of 6 months on and 6 months off, following protocols T1–T4. Fisetin is administered daily at a dose of 100 mg for the first two days of each month, repeated monthly, similarly to [43]. The results indicate that fisetin alone does not significantly improve E-efficacy. However, when combined with UC-II, a synergistic effect is observed, resulting in enhanced E-efficacy compared to UC-II treatment alone.

In Table 4, we present the computed E-efficacy at age 90 for individuals who began treatment at the ages indicated in the top horizontal row and discontinued treatment at ages 50, 55, 60, …, 85, and 90 (shown in the vertical column). UC-II is administered daily in 6-month cycles (6 months on, 6 months off), while fisetin is administered daily during the first two days of each month, repeated monthly.

Table 4 shows that combination of UC-II and fisetin is synergetic, in the sense that

$$[\text{Efficacy of (UC-II+fisetin)}] > [\text{Efficacy of UC-II}] + [\text{Efficacy of fisetin}]. \qquad (19)$$

Table 4 also shows the importance of starting treatment early, and continuing with it as long as possible. For example, if treatment began at age 50, the Efficacy achieved by age 70 is

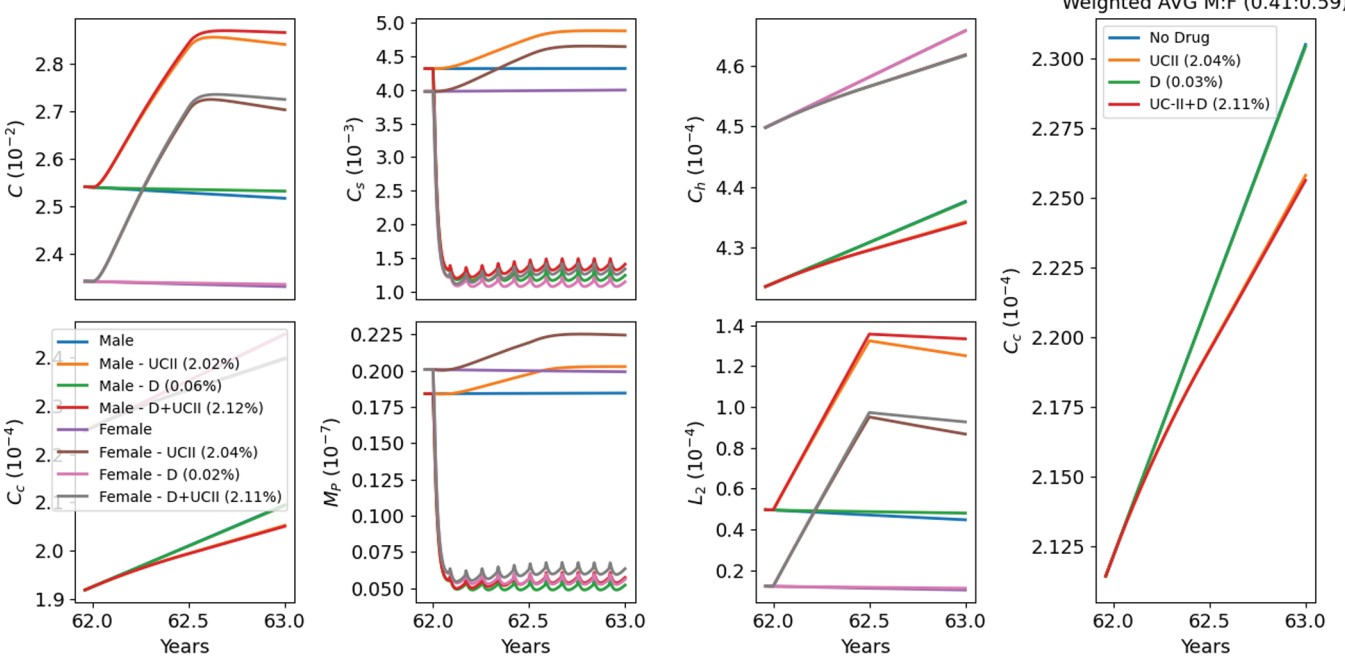

**Fig 8. Treatment with UC-II and fisetin.** UC-II is taken daily at a dose of 40 mg for a period of 6 months, followed by a discontinuation phase of 6 months. Fisetin (*D*) is taken during the first 2 days of the month, repeated each month.

**Table 4. Efficacy of $C_c$ reduction (in %) at year 90 in males and females with OA treated with UC-II and fisetin (D).** UC-II is administered daily for 6 months, followed by a 6-month break, in repeated cycle. Fisetin is taken during the first two days of each month and repeated monthly. Treatment begins at various times (Treatment Initiation) and ends at different times (Endtime).

(a) Males

| Years | | 50 | 55 | 60 | 65 | 70 | 75 | 80 | 85 | 90 | ← Initiation |
|---|---|---|---|---|---|---|---|---|---|---|---|
| 50 | UC-II | 0.00 | | | | | | | | | |
| | D | 0.00 | | | | | | | | | |
| | UC-II+D | 0.00 | | | | | | | | | |
| 55 | UC-II | 16.43 | 0.00 | | | | | | | | |
| | D | 17.87 | 0.00 | | | | | | | | |
| | UC-II+D | 53.77 | 0.00 | | | | | | | | |
| 60 | UC-II | 29.30 | 19.15 | 0.00 | | | | | | | |
| | D | 17.82 | 11.54 | 0.00 | | | | | | | |
| | UC-II+D | 72.97 | 51.35 | 0.00 | | | | | | | |
| 65 | UC-II | 41.14 | 32.97 | 21.40 | 0.00 | | | | | | |
| | D | 17.77 | 11.50 | 7.01 | 0.00 | | | | | | |
| | UC-II+D | 84.90 | 70.07 | 48.50 | 0.00 | | | | | | |
| 70 | UC-II | 52.46 | 45.20 | 35.74 | 22.93 | 0.00 | | | | | |
| | D | 17.73 | 11.47 | 6.98 | 3.98 | 0.00 | | | | | |
| | UC-II+D | 87.73 | 80.75 | 65.66 | 44.59 | 0.00 | | | | | |
| 75 | UC-II | 63.08 | 56.28 | 47.73 | 37.10 | 23.38 | 0.00 | | | | |
| | D | 17.69 | 11.44 | 6.96 | 3.96 | 2.07 | 0.00 | | | | |
| | UC-II+D | 91.88 | 84.51 | 74.55 | 59.27 | 39.15 | 0.00 | | | | |
| 80 | UC-II | 72.29 | 65.72 | 57.59 | 47.82 | 36.22 | 22.10 | 0.00 | | | |
| | D | 17.67 | 11.42 | 6.95 | 3.95 | 2.06 | 0.95 | 0.00 | | | |
| | UC-II+D | 94.03 | 86.71 | 77.68 | 65.96 | 50.55 | 31.79 | 0.00 | | | |
| 85 | UC-II | 78.84 | 72.38 | 64.43 | 54.99 | 44.10 | 31.77 | 17.90 | 0.00 | | |
| | D | 17.66 | 11.41 | 6.94 | 3.95 | 2.06 | 0.95 | 0.35 | 0.00 | | |
| | UC-II+D | 95.23 | 88.19 | 78.87 | 68.15 | 54.78 | 39.24 | 22.15 | 0.00 | | |
| 90 | UC-II | 81.39 | 74.95 | 67.04 | 57.69 | 46.95 | 34.97 | 22.04 | 9.04 | 0.00 | |
| | D | 17.65 | 11.41 | 6.94 | 3.94 | 2.06 | 0.95 | 0.35 | 0.07 | 0.00 | |
| | UC-II+D | 95.95 | 88.69 | 79.58 | 68.59 | 55.62 | 40.83 | 25.10 | 9.90 | 0.00 | |
| ↑ Endtime | | | | | | | | | | | |

(b) Females

| Years | | 50 | 55 | 60 | 65 | 70 | 75 | 80 | 85 | 90 | ← Initiation |
|---|---|---|---|---|---|---|---|---|---|---|---|
| 50 | UC-II | 0.00 | | | | | | | | | |
| | D | 0.00 | | | | | | | | | |
| | UC-II+D | 0.00 | | | | | | | | | |
| 55 | UC-II | 8.74 | 0.00 | | | | | | | | |
| | D | 4.97 | 0.00 | | | | | | | | |
| | UC-II+D | 23.71 | 0.00 | | | | | | | | |
| 60 | UC-II | 18.77 | 12.92 | 0.00 | | | | | | | |
| | D | 4.95 | 2.43 | 0.00 | | | | | | | |
| | UC-II+D | 42.70 | 29.93 | 0.00 | | | | | | | |
| 65 | UC-II | 29.95 | 24.83 | 16.47 | 0.00 | | | | | | |
| | D | 4.94 | 2.42 | 1.27 | 0.00 | | | | | | |
| | UC-II+D | 59.80 | 50.31 | 34.52 | 0.00 | | | | | | |
| 70 | UC-II | 41.57 | 36.71 | 29.43 | 19.03 | 0.00 | | | | | |
| | D | 4.93 | 2.41 | 1.27 | 0.70 | 0.00 | | | | | |
| | UC-II+D | 73.08 | 65.11 | 53.41 | 35.84 | 0.00 | | | | | |
| 75 | UC-II | 52.90 | 48.15 | 41.25 | 32.21 | 20.34 | 0.00 | | | | |
| | D | 4.93 | 2.41 | 1.26 | 0.70 | 0.39 | 0.00 | | | | |
| | UC-II+D | 82.17 | 74.90 | 64.88 | 51.72 | 33.83 | 0.00 | | | | |
| 80 | UC-II | 62.96 | 58.24 | 51.50 | 42.94 | 32.58 | 19.88 | 0.00 | | | |
| | D | 4.92 | 2.41 | 1.26 | 0.70 | 0.38 | 0.20 | 0.00 | | | |
| | UC-II+D | 87.65 | 80.69 | 71.37 | 59.78 | 45.86 | 28.68 | 0.00 | | | |
| 85 | UC-II | 70.24 | 65.54 | 58.85 | 50.46 | 40.58 | 29.30 | 16.53 | 0.00 | | |
| | D | 4.92 | 2.40 | 1.26 | 0.69 | 0.38 | 0.20 | 0.09 | 0.00 | | |
| | UC-II+D | 90.40 | 83.56 | 74.50 | 63.44 | 50.66 | 36.35 | 20.52 | 0.00 | | |
| 90 | UC-II | 73.08 | 68.39 | 61.71 | 53.36 | 43.57 | 32.56 | 20.60 | 8.49 | 0.00 | |
| | D | 4.92 | 2.40 | 1.26 | 0.69 | 0.38 | 0.20 | 0.09 | 0.02 | 0.00 | |
| | UC-II+D | 90.85 | 84.19 | 75.31 | 64.47 | 51.91 | 38.07 | 23.48 | 9.30 | 0.00 | |
| ↑ Endtime | | | | | | | | | | | |

87.73% for men and 73.06% for women, but if treatment began at age 60, then Efficacy by age 70 is only 65.66% for men and 53.41% for women.

In OA clinical trials [45], fisetin was administered in one experiment 2 days per month and in another experiment daily, for 90 days.

We can use the model to perform various other trials *in-silico* with more frequent injections of fisetin than in Table 4. We maintained the treatment protocol with UC-II as specified in Table 4, and evaluated the effect of administering fisetin at a dosage of 100 g/day with increased frequency. In Table 5, fisetin is administered during the first two days of each week, repeated weekly from the initiation of treatment until age 90.

In Table 5, the E-efficacy of fisetin alone (*D*), and consequently that of the combination UC-II+*D*, exceeds the efficacy observed in Table 4, where fisetin was administered only during the first two days of each month. For example, if the combined therapy began at age 50, the Efficacy achieved by age 70 was 89.43% for men and 74.28% for women, instead of 87.73% and 73.06% when fisetin was administered only the first two days of each month.

We can use the model to compute the Efficacy of other schedules of fisetin administration. For example, in Table 6 fisetin is administered daily for 7 days followed by 7 days off. This treatment is shown to result in very small improvement over Table 5 of Efficacy by age 90.

## 3 Conclusion

Osteoarthritis (OA) is the most prevalent form of joint disease, characterized by the progressive breakdown of articular cartilage and the associated joint pain, stiffness, and inflammation. The cartilage, a spongy tissue composed primarily of water, collagen, and chondrocytes, provides the mechanical buffer that facilitates pain-free joint motion. In OA, this protective tissue gradually deteriorates, and since cartilage has limited regenerative capacity, the damage is effectively irreversible. Understanding the biological progression and treatment dynamics of OA is essential for developing long-term management strategies, especially given its rising prevalence in aging populations.

In this study, we developed a mathematical model of osteoarthritis that captures key components of the cartilage structure in aging population. The model includes chondrocytes in their healthy, senescent, and hypertrophic states, along with calcified cartilage, matrix metalloproteinase-13 (MMP13), and collagen type II. The model was extended to simulate the effects of two therapeutic approaches: intra-articular injection of undenatured collagen type II (UC-II) and treatment with the senolytic drug fisetin. These therapies were evaluated individually and in combination to understand their potential synergy in mitigating disease progression.

We defined the concept of Efficacy of treatment for men and for women, and E-efficacy for a mixed group of men and women. We then demonstrated that the reduced pain associated with treatment by UC-II as documented in several clinical studies is proportional to the E-efficacy as computed by the mathematical model (Table 2).

In Tables 4, 5 and 6, we conducted a series of *in-silico* clinical studies to explore how the timing, duration, and frequency of treatment affected therapeutic outcomes. These simulations examined treatment initiation at various ages and tracked efficacy over multi-decade time horizons. In Table 4 we followed treatment schedules as in clinical studies, where UC-II was injected continuously for the first 6 months of each year and fisetin was injected in two consecutive days every 28 days. Further simulations explored alternative administration schedules, such as bi-weekly dosing during the first two days of each week (Table 5). These regimens demonstrated higher efficacy compared to monthly schedules, indicating that more frequent drug delivery can enhance therapeutic outcomes.

**Table 5. Efficacy of $C_c$ reduction (in %) at year 90 in males and females with OA treated with UC-II and fisetin (D).** UC-II is administered daily for 6 months, followed by a 6-month break, in repeated cycle. Fisetin is taken during the first two days of each week and repeated weekly. Treatment begins at various times (Treatment Initiation) and ends at various times (Endtime).

(a) Males

| Years | | 50 | 55 | 60 | 65 | 70 | 75 | 80 | 85 | 90 | ← Initiation |
|---|---|---|---|---|---|---|---|---|---|---|---|
| 50 | UC-II | 0.00 | | | | | | | | | |
| | D | 0.00 | | | | | | | | | |
| | UC-II+D | 0.00 | | | | | | | | | |
| 55 | UC-II | 16.43 | 0.00 | | | | | | | | |
| | D | 18.93 | 0.00 | | | | | | | | |
| | UC-II+D | 55.62 | 0.00 | | | | | | | | |
| 60 | UC-II | 29.30 | 19.15 | 0.00 | | | | | | | |
| | D | 18.87 | 12.18 | 0.00 | | | | | | | |
| | UC-II+D | 74.72 | 52.82 | 0.00 | | | | | | | |
| 65 | UC-II | 41.14 | 32.97 | 21.40 | 0.00 | | | | | | |
| | D | 18.81 | 12.13 | 7.37 | 0.00 | | | | | | |
| | UC-II+D | 86.10 | 71.41 | 49.62 | 0.00 | | | | | | |
| 70 | UC-II | 52.46 | 45.20 | 35.74 | 22.93 | 0.00 | | | | | |
| | D | 18.76 | 12.10 | 7.34 | 4.16 | 0.00 | | | | | |
| | UC-II+D | 89.43 | 81.79 | 66.63 | 45.39 | 0.00 | | | | | |
| 75 | UC-II | 63.08 | 56.28 | 47.73 | 37.10 | 23.38 | 0.00 | | | | |
| | D | 18.73 | 12.07 | 7.32 | 4.15 | 2.16 | 0.00 | | | | |
| | UC-II+D | 91.49 | 84.93 | 75.27 | 59.92 | 39.67 | 0.00 | | | | |
| 80 | UC-II | 72.29 | 65.72 | 57.59 | 47.82 | 36.22 | 22.10 | 0.00 | | | |
| | D | 18.70 | 12.05 | 7.30 | 4.14 | 2.15 | 0.99 | 0.00 | | | |
| | UC-II+D | 94.28 | 86.96 | 77.97 | 66.42 | 50.93 | 32.08 | 0.00 | | | |
| 85 | UC-II | 78.84 | 72.38 | 64.43 | 54.99 | 44.10 | 31.77 | 17.90 | 0.00 | | |
| | D | 18.68 | 12.03 | 7.29 | 4.13 | 2.14 | 0.98 | 0.36 | 0.00 | | |
| | UC-II+D | 95.32 | 88.37 | 79.00 | 68.34 | 55.05 | 39.43 | 22.27 | 0.00 | | |
| 90 | UC-II | 81.39 | 74.95 | 67.04 | 57.69 | 46.95 | 34.97 | 22.04 | 9.04 | 0.00 | |
| | D | 18.68 | 12.03 | 7.29 | 4.13 | 2.14 | 0.98 | 0.36 | 0.08 | 0.00 | |
| | UC-II+D | 96.01 | 88.82 | 79.69 | 68.75 | 55.84 | 40.98 | 25.18 | 9.92 | 0.00 | |
| ↑ Endtime | | | | | | | | | | | |

(b) Females

| Years | | 50 | 55 | 60 | 65 | 70 | 75 | 80 | 85 | 90 | ← Initiation |
|---|---|---|---|---|---|---|---|---|---|---|---|
| 50 | UC-II | 0.00 | | | | | | | | | |
| | D | 0.00 | | | | | | | | | |
| | UC-II+D | 0.00 | | | | | | | | | |
| 55 | UC-II | 8.74 | 0.00 | | | | | | | | |
| | D | 5.28 | 0.00 | | | | | | | | |
| | UC-II+D | 24.61 | 0.00 | | | | | | | | |
| 60 | UC-II | 18.77 | 12.92 | 0.00 | | | | | | | |
| | D | 5.26 | 2.57 | 0.00 | | | | | | | |
| | UC-II+D | 43.99 | 30.85 | 0.00 | | | | | | | |
| 65 | UC-II | 29.95 | 24.83 | 16.47 | 0.00 | | | | | | |
| | D | 5.24 | 2.56 | 1.34 | 0.00 | | | | | | |
| | UC-II+D | 61.17 | 51.49 | 35.38 | 0.00 | | | | | | |
| 70 | UC-II | 41.57 | 36.71 | 29.43 | 19.03 | 0.00 | | | | | |
| | D | 5.23 | 2.55 | 1.33 | 0.73 | 0.00 | | | | | |
| | UC-II+D | 74.28 | 66.18 | 54.34 | 36.53 | 0.00 | | | | | |
| 75 | UC-II | 52.90 | 48.15 | 41.25 | 32.21 | 20.34 | 0.00 | | | | |
| | D | 5.22 | 2.54 | 1.33 | 0.73 | 0.40 | 0.00 | | | | |
| | UC-II+D | 83.11 | 75.74 | 65.63 | 52.36 | 34.30 | 0.00 | | | | |
| 80 | UC-II | 62.96 | 58.24 | 51.50 | 42.94 | 32.58 | 19.88 | 0.00 | | | |
| | D | 5.22 | 2.54 | 1.32 | 0.73 | 0.40 | 0.21 | 0.00 | | | |
| | UC-II+D | 88.36 | 81.32 | 71.93 | 60.25 | 46.24 | 28.95 | 0.00 | | | |
| 85 | UC-II | 70.24 | 65.54 | 58.85 | 50.46 | 40.58 | 29.30 | 16.53 | 0.00 | | |
| | D | 5.21 | 2.54 | 1.32 | 0.73 | 0.40 | 0.21 | 0.09 | 0.00 | | |
| | UC-II+D | 90.78 | 84.05 | 74.93 | 63.79 | 50.92 | 36.54 | 20.63 | 0.00 | | |
| 90 | UC-II | 73.08 | 68.39 | 61.71 | 53.36 | 43.57 | 32.56 | 20.60 | 8.49 | 0.00 | |
| | D | 5.21 | 2.54 | 1.32 | 0.72 | 0.40 | 0.21 | 0.09 | 0.02 | 0.00 | |
| | UC-II+D | 91.27 | 84.51 | 75.59 | 64.75 | 52.13 | 38.22 | 23.56 | 9.32 | 0.00 | |
| ↑ Endtime | | | | | | | | | | | |

**Table 6. Efficacy of $C_c$ reduction (in %) at year 90 in males and females with OA treated with UC-II and fisetin (D).** UC-II is administered daily for 6 months, followed by a 6-month break, in repeated cycle. Fisetin is taken daily for 7 days, followed by a 7-day break, in a repeating cycle. Treatment begins at various times (Treatment Initiation) and ends at year 90 (Endtime).

(a) Males

| Years | | 50 | 55 | 60 | 65 | 70 | 75 | 80 | 85 | 90 | ← Initiation |
|---|---|---|---|---|---|---|---|---|---|---|---|
| 90 | UC-II | 81.39 | 74.95 | 67.04 | 57.69 | 46.95 | 34.97 | 22.04 | 9.04 | 0.00 | |
| | D | 18.78 | 12.09 | 7.32 | 4.14 | 2.15 | 0.98 | 0.36 | 0.08 | 0.00 | |
| | UC-II+D | 95.98 | 88.84 | 79.79 | 68.69 | 55.88 | 41.02 | 25.21 | 9.93 | 0.00 | |
| ↑ Endtime | | | | | | | | | | | |

(b) Females

| Years | | 50 | 55 | 60 | 65 | 70 | 75 | 80 | 85 | 90 | ← Initiation |
|---|---|---|---|---|---|---|---|---|---|---|---|
| 90 | UC-II | 73.08 | 68.39 | 61.71 | 53.36 | 43.57 | 32.56 | 20.60 | 8.49 | 0.00 | |
| | D | 5.20 | 2.54 | 1.33 | 0.73 | 0.40 | 0.21 | 0.09 | 0.02 | 0.00 | |
| | UC-II+D | 91.37 | 84.60 | 75.64 | 64.83 | 52.19 | 38.26 | 23.58 | 9.33 | 0.00 | |
| ↑ Endtime | | | | | | | | | | | |

The main results of this paper are:

(i) Efficacy of treatment with UC-II, as derived by the model, is positively correlated to the severity of OA.

(ii) Early and continuous treatment significantly improved efficacy, particularly when treatment initiated before age 60; delayed treatment initiation led to notably diminished outcomes.

(iii) Efficacy of the combination therapy of UC-II with fisetin consistently exceeded the sum of the treatment Efficacy by UC-II alone and by fisetin alone, highlighting the potential for drug synergy in OA management.

These results reinforce the importance of early treatment and choice of treatment protocol to achieve maximum clinical benefits.

The model has several limitations.

1. OA is a complex disease involving cartilage degradation, subchondral bone changes, synovial inflammation, and pain mechanisms independent of the structure damage. Accordingly, the severity of OA has been measured by different criteria. In this paper we consider aging patients, above 50 years, for whom the disease is mostly non-inflammatory [8], and we focus on the destruction of the cartilage, measuring the severity of OA in terms of cartilage calcification as in [24]. Cartilage calcification is promoted by hypertrophic chondrocytes, which in health are restricted to the subchondral region. We tacitly assumed that in OA subchondral bone changes result in constant transition from chondrocytes to hypertrophic chondrocytes. In health, the level of hypertrophic chondrocytes is controlled by collagen type II, but in OA the level of collagen type II is highly diminished, due to senescence in the aging patients, hence the concentration of hypertrophic chondrocytes keeps increasing and, correspondingly, also cartilage calcification. We also assumed that pain mechanisms independent of structural damage does not affect the cartilage calcification process.

2. Most of the model parameters are either taken from published biological data, or estimated using such data. However, some parameters (listed "this work" in Table 7) are unknown, including arguably the most important parameter $\lambda_{C_h C_c}$, which determines

**Table 7. Parameters for the model.**

| Parameters | Descriptions | Values | references |
|---|---|---|---|
| $\lambda_{CC_h}$ | rate of $C \to C_h$ transition | $3.26 \times 10^{-2}$ d$^{-1}$ | [62]est. |
| $\lambda_{CC_s}$ | rate of $C \to C_s$ transition | $6 \times 10^{-3}$ d$^{-1}$ | [10]est. |
| $\lambda_{C_hC_c}$ | rate of $C_c$ formation | $10^4$ d$^{-1}$ | this work |
| $\lambda_{C_sM_P}$ | rate of production of $M_P$ by $C_s$ | $6.013 \times 10^{-5}$ d$^{-1}$ | est. |
| $\lambda_{CL_2}$ | rate of production of $L_2$ by $C$ | $1.87 \times 10^{-8}$ d$^{-1}$ | est. |
| $\mu_C$ | death rate of $C$ | 0.03 d$^{-1}$ | [62] |
| $\mu_{C_s}$ | death rate of $C_s$ at $t = 0$ | 0.04 d$^{-1}$ | [35,62]est. |
| $\gamma_D$ | dose of fisetin | $1.52 \times 10^{-6}$ g/cm$^3$ d$^{-1}$ | [54][55] |
| $\gamma_{U_2}$ | dose of $U_2$ | $6.1 \times 10^{-7}$ g/cm$^3$ d$^{-1}$ | [37,38] |
| $\mu_{L_2M_P}$ | elimination rate of $M_P$ by $L_2$ | $4.897 \times 10^4$ cm$^3$/g d$^{-1}$ | est. |
| $\mu_{M_PL_2}$ | elimination rate of $L_2$ by $M_P$ | 100.2 cm$^3$/g d$^{-1}$ | est. |
| $\mu_{DC_s}$ | absorption rate of $D$ by $C_s$ | $3 \times 10^2$ cm$^3$/g d$^{-1}$ | this work |
| $\mu_{C_sD}$ | elimination rate of $C_s$ by $D$ | $10^3$ cm$^3$/g d$^{-1}$ | this work |
| $\delta_C$ | diffusion rate of $C$, $C_s$ and $C_h$ | $8.64 \times 10^{-7}$ cm$^2$ d$^{-1}$ | [52]est. |
| $\delta_{C_c}$ | diffusion rate of $C_c$ | 0 cm$^2$ d$^{-1}$ | |
| $\delta_{M_P}$ | diffusion rate of $M_P$ | $6.6 \times 10^{-2}$ cm$^2$ d$^{-1}$ | [49–51]est. |
| $\delta_{L_2}$ | diffusion rate of $L_2$ | $4.78 \times 10^{-2}$ cm$^2$ d$^{-1}$ | [49–51]est. |
| $A$ | source of chondrocytes | $1.41 \times 10^{-3}$ g/cm$^3$ d$^{-1}$ | [62]est. |
| $K_{L_2}$ | half-saturation of $L_2$ | $1.7 \times 10^{-4}$ g/cm$^3$ | [59,60]est. |
| $T_{wM_P}, T_{wL_2}$ | time-scale in postmenopausal effect cartilage | 12 years | this work |
| $\alpha_D$ | factor of effect of $D$ | $2 \times 10^6$ | this work |
| $\alpha_{U_2}$ | factor of effect of $U_2$ | 0.05 | this work |
| $\alpha_{wM_P}$ | rate of postmenopausal effect on $M_P$ | 2.5 d$^{-1}$ | this work |
| $\alpha_{wL_2}$ | rate of postmenopausal effect on $L_2$ | $2.5 \times 10^{-4}$ d$^{-1}$ | this work |
| $h$ | width of articular cartilage | 0.2 cm | [48] |

the rate of transition from chondrocyte hyperthophy to cartilage calcification. We performed sensitivity analysis in Fig 9, demonstrating that changes of ±50% of parameters result in expected changes in cartilage calcification. Hence, model simulations can be useful as qualitative results, but only in restricted range (±50%) about the chosen parameters.

3. Key parameters like drug effectiveness coefficients are not known, as those indicated in Table 7 by "this work". Hence the results of clinical studies in Tables 4–6 lack experimental validation. When these parameters are revised based on new experiments, the corresponding revised Tables 4–6 would become predictive.

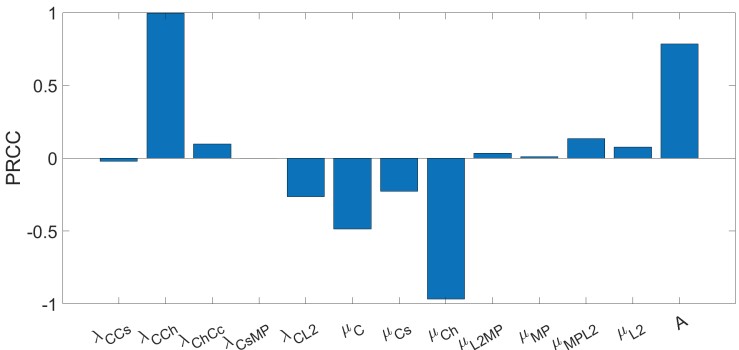

**Fig 9. Parameter sensitivity analysis for the calcified cartilage, $C_c$, after 1 year.** The p-values are less than 0.01.

The model developed in this paper, while comprehensive, simplifies complex biological processes and does not account for all systemic factors involved in OA or drug metabolism. Patient heterogeneity beyond sex and age – such as genetic variability, comorbid conditions, and lifestyle – was not included but could significantly impact treatment outcomes. Furthermore, our predictions are constrained by the accuracy of parameter estimates and assumptions within the model; in particular, the assumption that progression state of OA is measured simply by the density of calcified cartilage. The present paper can serve as a first step in future work that will involve refining these aspects, incorporating patient-specific data, and validating predictions through experimental and clinical collaborations.

## 4 Parameter sensitivity analysis

We performed sensitivity analysis with respect to the calcified cartilage density after 1 year, for a male OA patient, for the parameters $\lambda_{CC_s}$, $\lambda_{CC_h}$, $\lambda_{C_hC_c}$, $\lambda_{C_sM_P}$, $\lambda_{CL_2}$, $\mu_C$, $\mu_{C_s}$, $\mu_{C_h}$, $\mu_{L_2M_P}$, $\mu_{M_P}$, $\mu_{M_PL_2}$, $\mu_{L_2}$, and $A$.

The computations were done using Latin Hypercube Sampling/Partial Rank Correlation Coefficient (LHS/PRCC) with a Matlab package by [46,47]. The range for the parameters in the sensitivity analysis was between ±50% of their baseline values in Table 7.

The system dynamics reveal several key relationships governing calcified cartilage density ($C_c$). The baseline transition rate from chondrocytes to hypertrophic chondrocytes ($\lambda_{CC_h}$) demonstrates the strongest positive correlation with $C_c$, as it directly controls the production flux through $\partial C_h/\partial t$. While the calcified cartilage production rate ($\lambda_{C_hC_c}$) shows expected positive correlation, its effect is secondary to $\lambda_{CC_h}$ due to this hierarchical dependence. The chondrocyte production rate ($A$) also significantly influences $C_c$ by determining the progenitor population available for differentiation.

Conversely, the elimination rate of hypertrophic chondrocytes ($\mu_{C_h}$) produces the most pronounced negative effect on $C_c$ accumulation. Similar depletion effects occur through increased chondrocyte death rate ($\mu_C$), which reduces the source population. Additional negative correlations emerge with elevated elimination of senescent chondrocytes ($\mu_{C_s}$) and increased collagen type II secretion ($\lambda_{CL_2}$). These relationships collectively demonstrate that $C_c$ accumulation depends fundamentally on the balance between chondrocyte production ($A$) and differentiation ($\lambda_{CC_h}$) versus elimination processes ($\mu_C$, $\mu_{C_h}$, $\mu_{C_s}$).

## 5 Parameter estimations

The average thickness (width) of the femoral articular cartilage is $h$ = 1.98 mm [48].

### Estimates for the diffusion coefficients $\delta_X$

**Diffusion coefficients of MMP and collagen type II.** Young [49] established the following formula for estimating the diffusion coefficient $\delta_p$ of protein $p$:

$$\delta_p = \frac{M_V^{1/3}}{M_p^{1/3}}\delta_V, \tag{20}$$

where $M_V$ and $\delta_V$ are respectively the molecular weight and diffusion coefficient of VEGF, $M_p$ is the molecular weight of $p$, $M_V$ = 24kDa [50] and $\delta_V = 8.64 \times 10^{-2}$ cm$^2$ d$^{-1}$ [51].

The molecular weight of MMP-13 is 53,820 Da, and of collagen type II (COL2A1) is 141,785 Da [50]. Hence,

$$\delta_{M_P} = 6.6 \times 10^{-2} \text{ cm}^2 \text{ d}^{-1}, \quad \delta_{L_2} = 4.78 \times 10^{-2} \text{ cm}^2 \text{ d}^{-1}.$$

**Diffusion coefficients of chondrocytes and calcified cartilage.** We take

$$\delta_C = \delta_{C_h} = \delta_{C_s} = 8.64 \times 10^{-7} \text{ cm}^2 \text{ d}^{-1},$$

the same as for macrophages [52].

The calcified cartilage is hard bone-like connective tissue [53], so we take

$$\delta_{C_c} = 0.$$

## Drugs

Fisetin was used in clinical trials at dose of 100 mg/day [43].

The adult human body average volume is approximately $66 \times 10^3$ g/cm³ [54,55]. Hence the average dose density of fisetin in the body is

$$\gamma_D = \frac{100 \times 10^{-3} \text{ g d}^{-1}}{6.6 \times 10^4 \text{ cm}^3} = 1.52 \times 10^{-6} \text{ g/cm}^3 \text{ d}^{-1}.$$

The terminal half-life of fisetin is approximately 3 hours [56]. Hence

$$\mu_D = \frac{\ln 2}{0.125} = 5.55 \text{ d}^{-1}.$$

We take

$$\omega_D = 0.01 \text{ d}^{-1}.$$

The standard daily dose of UC-II is 40 mg [37,38]. Hence

$$\gamma_{U_2} = \frac{40 \times 10^{-3} \text{ g d}^{-1}}{6.6 \times 10^4 \text{ cm}^3} = 6.1 \times 10^{-7} \text{ g/cm}^3 \text{ d}^{-1}.$$

## Estimates by equations

In what follows, we shall use the "steady state" equations of the model to estimate some of the unknown parameters; this means the following:

We set the right-hand side of an equation to zero, and take all species $X$ (cells and proteins) in this equation, at their "average" values $X = X^0$ in OA of the control case (no drugs); for example, $C_s^0 = \frac{15}{100} C^0$.

**Eq (3):** The lifespan of hypertrophic chondrocytes is approximately 1 day [57,58]. We take the half-life to be $t_{C_h}^{1/2} = 0.25$ day. Hence,

$$\mu_{C_h} = \frac{\ln 2}{0.25} = 2.77 \text{ d}^{-1}.$$

The level of collagen type II in normal healthy femoral cartilage ranges between 139.9–179.5 $\mu$g/ml [59] (Table VII), and it degrades, on the average, by about 6 times in OA at age

>70 [60] (Table 1). We take $L_2 = 1.7 \times 10^{-4}$ g/cm$^3$ in health,

$$L_2 = \frac{1.7 \times 10^{-4}}{6} = 2.83 \times 10^{-5} \text{ g/cm}^3 \text{ in OA,}$$

and

$$K_{L_2} = 1.7 \times 10^{-4} \text{ g/cm}^3.$$

In health there is only a small amount $C_h$ [61].

We take $C_h^0 = 1\% C^0$, where $C^0 = 0.027$ g/cm$^3$ by [62]; hence $C_h^0 = 2.7 \times 10^{-4}$ g/cm$^3$. Assuming a nearly steady state of Eq (3), we have

$$\lambda_{CC_h} C^0 \frac{1}{1 + L_2/K_{L_2}} - \mu_{C_h} C_h^0 \simeq 0, \text{ or equivalently, } \lambda_{CC_h} \simeq \frac{\mu_{C_h} C_h^0 (1 + L_2/K_{L_2})}{C^0},$$

where $\mu_{C_h} = 2.77$ d$^{-1}$. Hence,

$$\lambda_{CC_h} = 3.26 \times 10^{-2} \text{ d}^{-1}.$$

**Eq (2):** Senescent cells have a relatively short half-life, presumably because they are efficiently cleared by immune cells [35]. With $\mu_C = 0.03$ d$^{-1}$ [62], we take

$$\mu_{C_s}(0) = 0.04 \text{ d}^{-1},$$

and, by Eq (5),

$$\mu_{C_s}(t) = \frac{0.04}{1 + t/(90 \text{ years})} \text{ d}^{-1}.$$

From the average steady state of Eq (2), we get

$$\lambda_{CC_s} = \mu_{C_s} \frac{C_s}{C}.$$

Between 30 and 70 years of age there is a fall of 30% from $C$ to $C_s$ [10]. We assume

$$\frac{C_s}{C} \simeq \frac{15}{100} \text{ on average.}$$

Hence,

$$\lambda_{CC_s} = 0.15 \times \mu_{C_s} = 6 \times 10^{-3} \text{ d}^{-1}.$$

From steady state of Eq (1), $A - \lambda_{CC_s} C - \lambda_{CC_h} C(\frac{1}{2}) - \mu_C C = 0$, where $\mu_C = 0.03$ d$^{-1}$, $C^0 = 0.027$ g/cm$^3$. Hence

$$A = 1.41 \times 10^{-3} \text{ g/cm}^3 \text{ d}^{-1}.$$

**Eqs (6) and (7):** The half-life of MMP-13 is approximately 70 minutes (0.05 days) [63]. Hence,

$$\mu_{M_P} = \frac{\ln 2}{0.05} = 13.86 \text{ d}^{-1}.$$

The average level of MMP in the synovial fluid is 16.17 ng/ml in rheumatoid arthritis (RA) patient and 0.75 ng/ml in health [64]. We assume that the level of MMP in OA is the same as in RA and take

$$M_P = 16.17 \times 10^{-9} \text{ g/cm}^3 \text{ in OA.}$$

The half-life of cartilage collagen type II is approximately 117 years (42,705 days) [65]. Hence,

$$\mu_{L_2} = \frac{\ln 2}{42705} = 1.62 \times 10^{-5} \text{ d}^{-1}.$$

In steady state of Eq (7), we have $\lambda_{CL_2} C^0 - \mu_{MpL_2} L_2^0 M_P^0 - \mu_{L_2} L_2^0 = 0$, with $C^0 = 0.027$ g/cm$^3$, $L_2^0 = 2.83 \times 10^{-5}$ g/cm$^3$, $\mu_{L_2} = 1.62 \times 10^{-5}$ d$^{-1}$, and $M_P^0 = 16.17 \times 10^{-9}$ g/cm$^3$ in OA. We assume that the loss rate of active $L_2$ by $M_P$ is smaller than the degradation rate of $L_2$, and take

$$10\mu_{MpL_2} M_P^0 = \mu_{L_2}, \text{ so that } \mu_{MpL_2} = 100.2 \text{cm}^3/\text{g d}^{-1}.$$

Then

$$\lambda_{CL_2} = 1.87 \times 10^{-8} \text{ d}^{-1}.$$

In steady state of Eq (6), we have $\lambda_{C_sM_P} C_s^0 - \mu_{L_2M_P} M_P^0 L_2^0 - \mu_{M_P} M_P^0 = 0$, with $C_s^0 = 0.15C^0 = 0.15 \times 0.027 = 4.1 \times 10^{-3}$ g/cm$^3$, $\mu_{M_P} = 13.86$ d$^{-1}$. We assume that the rate by which $M_P$ is decreased by blocking $L_2$ is smaller than the degradation rate of $M_P$ and take

$$10\mu_{L_2M_P} L_2^0 = \mu_{M_P}, \text{ so that } \mu_{L_2M_P} = 4.897 \times 10^4 \text{cm}^3/\text{g d}^{-1}.$$

Then

$$\lambda_{C_sM_P} = 6.013 \times 10^{-5} \text{ d}^{-1}.$$

## Author contributions

**Conceptualization:** Nourridine Siewe, Avner Friedman.

**Data curation:** Nourridine Siewe, Avner Friedman.

**Formal analysis:** Nourridine Siewe, Avner Friedman.

**Funding acquisition:** Nourridine Siewe, Avner Friedman.

**Investigation:** Nourridine Siewe, Avner Friedman.

**Methodology:** Nourridine Siewe, Avner Friedman.

**Project administration:** Nourridine Siewe, Avner Friedman.

**Resources:** Nourridine Siewe, Avner Friedman.

**Software:** Nourridine Siewe.

**Supervision:** Nourridine Siewe, Avner Friedman.

**Validation:** Nourridine Siewe, Avner Friedman.

**Visualization:** Nourridine Siewe, Avner Friedman.

**Writing – original draft:** Nourridine Siewe, Avner Friedman.

**Writing – review & editing:** Nourridine Siewe, Avner Friedman.

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
