## [Decision Letter · Decision Letter 0]

6 Aug 2025

PONE-D-25-32495Modeling treatment of osteoarthritis with standard therapy and senolytic drugsPLOS ONE

Dear Dr. Siewe,

Thank you for submitting your manuscript to PLOS ONE. After careful consideration, we feel that it has merit but does not fully meet PLOS ONE’s publication criteria as it currently stands. Therefore, we invite you to submit a revised version of the manuscript that addresses the points raised during the review process.

We look forward to receiving your revised manuscript.

Kind regards,

Xindie Zhou

Academic Editor

PLOS ONE

Journal Requirements:

4. Thank you for stating the following financial disclosure: [Research reported in this publication was supported by the National Institute Of General Medical Sciences of the National Institutes of Health under Award Number R16GM154782. The content is solely the responsibility of the authors and does not necessarily represent the official views of the National Institutes of Health.]. 

5. We note that your Data Availability Statement is currently as follows: [All relevant data are within the manuscript and its Supporting Information files.]

6. Thank you for stating the following in the Acknowledgments Section of your manuscript: [Research reported in this publication was supported by the National Institute Of General

Medical Sciences of the National Institutes of Health under Award Number R16GM154782.

The content is solely the responsibility of the authors and does not necessarily represent the

official views of the National Institutes of Health.]

Please remove any funding-related text from the manuscript and let us know how you would like to update your Funding Statement. Currently, your Funding Statement reads as follows: [Research reported in this publication was supported by the National Institute Of General Medical Sciences of the National Institutes of Health under Award Number R16GM154782. The content is solely the responsibility of the authors and does not necessarily represent the official views of the National Institutes of Health.]. 

Reviewers' comments:

Reviewer's Responses to Questions

**Comments to the Author**

1. Is the manuscript technically sound, and do the data support the conclusions?

Reviewer #1: Yes

Reviewer #2: Yes

Reviewer #3: Yes

2. Has the statistical analysis been performed appropriately and rigorously? 

Reviewer #1: Yes

Reviewer #2: Yes

Reviewer #3: Yes

3. Have the authors made all data underlying the findings in their manuscript fully available?

Reviewer #1: Yes

Reviewer #2: Yes

Reviewer #3: Yes

4. Is the manuscript presented in an intelligible fashion and written in standard English?

Reviewer #1: Yes

Reviewer #2: Yes

Reviewer #3: Yes

5. Review Comments to the Author

Reviewer #1: This review highlights the manuscript's ambitious attempt to model a complex disease but identifies several critical issues that would need to be addressed before publication. The primary concerns center on model validation, parameter uncertainty, and the disparity between the mathematical and clinical applicability of the results.

The authors have undertaken a challenging modeling problem, but the work would benefit significantly from more rigorous validation approaches and better integration with clinical outcomes. The very low efficacy values (0.15-0.22%) compared to substantial clinical improvements (22-33% reduction in WOMAC scores) suggest that the model may not be capturing the most clinically relevant aspects of OA progression and treatment response.

Summary

This manuscript presents a mathematical model of osteoarthritis (OA) progression that incorporates cellular senescence and treatment effects of undenatured collagen type II (UC-II) and the senolytic drug fisetin. The authors develop separate models for males and females, validate them against clinical trial data, and conduct in silico trials to assess treatment efficacy and synergy.

Novel Integration: The incorporation of cellular senescence mechanisms into OA modeling is timely and biologically relevant, addressing a significant gap in current therapeutic approaches.

Sex-Specific Modeling: The differentiation between male and female models, particularly accounting for postmenopausal effects on collagen degradation and MMP-13 levels, is a valuable contribution.

Clinical Validation: The authors attempt to validate their model against four published clinical trials (T1-T4), demonstrating a correlation between computed E-efficacy and reported improvements in WOMAC scores. Extensive parameter estimation based on literature values and steady-state assumptions demonstrates thorough model development.

Major Concerns

1. Model Assumptions and Simplifications

Critical Issue: The fundamental assumption that OA progression is solely measured by calcified cartilage density (Cc) is oversimplified and potentially problematic. OA is a complex, multifactorial disease involving:

Cartilage degradation beyond calcification, Synovial inflammation, Subchondral bone changes, and Pain mechanisms independent of structural damage. However, authors acknowledge this limitation but don't adequately address how this affects model validity.

The model contains numerous parameters (>25) estimated from disparate sources, steady-state assumptions, and author assumptions; however, no sensitivity analysis is provided. For example, the choice of λ_{Ch, Cc} = 10^4 d^{-1} appears arbitrary without justification.

The correlation between E-efficacy and WOMAC scores shows very low efficacy values (0.15-0.22%) for treatments that demonstrated improvements of 22-33% in WOMAC scores. This raises questions about the model's predictive validity and whether it captures meaningful therapeutic effects.

Furthermore, the relationship between senescent cell elimination and improvement in OA progression is assumed rather than established. Clinical trial data for fisetin show minimal effects, yet the model predicts synergy. The correlation between model predictions and clinical outcomes is weak.

Minor Issues

The authors claim model validation based on four trials, but the correlation is weak.

Treatment protocols in Tables 4-6 don't match standard clinical practice, and the 6-month on/off UC-II protocol lacks clinical justification.

No confidence intervals or uncertainty quantification provided.

No statistical comparison between treatment groups

Key parameters like drug effectiveness coefficients lack experimental validation

Reviewer #2: I have extensively read through the manuscript titled, "Modeling treatment of osteoarthritis with standard therapy and senolytic drugs", and I am highly impressed with the presentation of the work. The manuscript presents a comprehensive mathematical model of osteoarthritis (OA) incorporating biological mechanisms such as chondrocyte senescence, hypertrophy, calcified cartilage formation, and the roles of collagen type II and MMP13. It evaluates treatment effects using UC-II and senolytic drugs (fisetin), both independently and in combination, providing a novel in-silico framework validated against clinical trial data. The work is well-structured, technically rigorous, and contributes valuable computational insight into OA treatment planning.

However, here are some few suggested issues related to the work to improve it:

1. The study uses calcified cartilage (Cc) as a proxy for OA severity and compares it to WOMAC pain scores. The authors should explicitly discuss the biological or clinical rationale for using Cc density as a surrogate for clinical pain outcomes, even if only qualitatively.

2. While the mixed-type boundary condition is mathematically acceptable, more biological interpretation or justification would strengthen the modeling assumptions, especially for protein/drug diffusion.

3. Consider including a sensitivity analysis to identify which parameters have the greatest influence on model outcomes, particularly on efficacy metrics. Even a local sensitivity analysis would improve robustness.

4. In some places, there are unclear or potentially redundant terms in equations (e.g., repeated ∂Ch/∂t in Eq. 2.3). Carefully review all equations for typographical consistency.

5. Ensure that all variables shown in the schematic diagram (Figure 1) are labeled and defined in the caption or immediately adjacent text.

6. Define all symbols the first time they appear (e.g., KL2 in Eq. 2.1), even if they are listed in Table 1. This will improve readability for interdisciplinary readers.

Reviewer #3: The article its good and the results it's correct so we can accept this article with minor correction

Minor correction

The Authors must add recent references

10.3934/MATH.2023821

10.3390/sym15020286

10.1155/2022/2138165

10.1155/2022/5468696

6. PLOS authors have the option to publish the peer review history of their article (what does this mean?). If published, this will include your full peer review and any attached files.

Reviewer #1: No

Reviewer #2: No

Reviewer #3: No

---

## [Author Response · Author response to Decision Letter 1]

19 Aug 2025

See Response_to_Reviewer.pdf file

---

## [Decision Letter · Decision Letter 1]

4 Sep 2025

Modeling treatment of osteoarthritis with standard therapy and senolytic drugs

PONE-D-25-32495R1

Dear Dr. Siewe,

We’re pleased to inform you that your manuscript has been judged scientifically suitable for publication and will be formally accepted for publication once it meets all outstanding technical requirements.

Kind regards,

Xindie Zhou

Academic Editor

PLOS ONE

Additional Editor Comments (optional):

Reviewer #1:

Reviewer #2:

Reviewers' comments:

Reviewer's Responses to Questions

**Comments to the Author**

1. If the authors have adequately addressed your comments raised in a previous round of review and you feel that this manuscript is now acceptable for publication, you may indicate that here to bypass the “Comments to the Author” section, enter your conflict of interest statement in the “Confidential to Editor” section, and submit your "Accept" recommendation.

Reviewer #1: All comments have been addressed

Reviewer #2: All comments have been addressed

2. Is the manuscript technically sound, and do the data support the conclusions?

Reviewer #1: Yes

Reviewer #2: Yes

3. Has the statistical analysis been performed appropriately and rigorously? 

Reviewer #1: I Don't Know

Reviewer #2: Yes

4. Have the authors made all data underlying the findings in their manuscript fully available?

Reviewer #1: Yes

Reviewer #2: No

5. Is the manuscript presented in an intelligible fashion and written in standard English?

Reviewer #1: Yes

Reviewer #2: Yes

6. Review Comments to the Author

Reviewer #1: The authors don't address how clinicians or researchers should interpret or apply model predictions given the weak correlation with established clinical measures.

The work shows promise as a foundation for OA modeling, but requires additional development to demonstrate its value for clinical decision-making or therapeutic development.

Reviewer #2: (No Response)

7. PLOS authors have the option to publish the peer review history of their article (what does this mean?). If published, this will include your full peer review and any attached files.

Reviewer #1: No

Reviewer #2: No

---

## [Editor Report · Acceptance letter]

PONE-D-25-32495R1

PLOS ONE

Dear Dr. Siewe,

I'm pleased to inform you that your manuscript has been deemed suitable for publication in PLOS ONE. Congratulations! Your manuscript is now being handed over to our production team.

Kind regards,

on behalf of

Dr. Xindie Zhou

Academic Editor

PLOS ONE